# The Relationship between Dog Ownership, Psychopathological Symptoms and Health-Benefitting Factors in Occupations at Risk for Traumatization

**DOI:** 10.3390/ijerph17072562

**Published:** 2020-04-08

**Authors:** Johanna Lass-Hennemann, Sarah K. Schäfer, M. Roxanne Sopp, Tanja Michael

**Affiliations:** Department of Psychology, Saarland University, 66123 Saarbrücken, Germany; sarah.schaefer@uni-saarland.de (S.K.S.); roxanne.sopp@uni-saarland.de (M.R.S.); t.michael@mx.uni-saarland.de (T.M.)

**Keywords:** mental health, pet ownership, high-risk occupation, PTSD, posttraumatic stress, burnout, sense of coherence, resilience, locus of control, dog

## Abstract

Individuals working in high-risk occupations (e.g., emergency staff) are exposed to high levels of occupational stress including traumatic events. Correspondingly, several studies report high rates of mental health problems among these occupations. Pet ownership has been associated with better mental health. However, to date a study on the association between pet ownership and indicators of mental health in these occupations is missing. The present cross-sectional survey (*N* = 580) investigated pet ownership, attachment to pets, health-benefitting factors (i.e., sense of coherence, trait-resilience, locus of control) and psychopathological symptoms (i.e., general mental health problems, posttraumatic stress, burnout) in medical staff, police officers, and firefighters. Dog owners and non-dog owners showed comparable levels of psychopathological distress and health-benefitting factors. Compared to cat owners, dog owners demonstrated stronger emotional attachment to their pet. Moreover, a stronger attachment was also linked to higher levels of psychopathological symptoms and lower levels of health-benefitting factors. However, the relationship between attachment to pets and health-benefitting factors could be explained by their overlap with psychopathological symptom levels. Overall, our findings are not in line with the notion that pet ownership generally has a health-benefitting effect. Future studies need to investigate circumstances that modulate positive effects of pet ownership.

## 1. Introduction

Some occupations are not only exposed to considerable levels of occupational stress but are also constantly at a high risk for experiencing traumatic events. While approximately 70% of the global civilian population report to experience of at least one traumatic event during their lifetime [1,2], this proportion increases to 84% for those working in high-risk occupations (e.g., police officers, firefighters and emergency dispatchers [3]). In particular, individuals working in these occupations are exposed to work-related traumatic events repeatedly resulting in a cumulative burden which increases their risk for developing mental health problems such as burnout, depressive symptoms and posttraumatic stress disorder (PTSD).

However, responses to these cumulative stressors differ significantly across employees. While some develop mental health problems, others are able to maintain their mental health even when faced with persisting stressful circumstances [4,5,6]. Based on these diverging responses, it is of major importance to identify factors relevant to successful coping.

In that regard, pet ownership has been linked to enhanced physical and mental health in the general population as well as in patients with physical and mental disorders (for a review see Wells et al. [7]). For instance, pets have been shown to alleviate the perception of loneliness and depression and to improve perceived general health in older populations [8]. However, at the same time, some studies also find null or even negative effects of pet ownership on physical and psychological health and raise the question whether pet ownership can be robustly associated with better health (for a critical review, see Herzog [9]).

Despite these divergent findings, human-animal interactions have been consistently shown to reduce subjective and physiological stress levels (for a meta-analysis, see Ein et al. [10]), function as social support and enhance social interactions (for a review, see Beetz et al. [11]) and to decrease depressive symptoms (for meta-analyses, see Borgi et al. [12], and Souter and Miller [13]). This leads to the hypothesis that pet ownership might also have beneficial effects on individuals working in high-risk occupations. To date, there are only a few studies examining the role of pet ownership in responses to stressors and critical life events. Two studies investigated pet ownership after a social loss, showing that bereaved individuals owning a pet were less depressed, felt less lonely and reported less use of medication than those without pets [14,15]. Moreover, a study on war-traumatized children showed that those with a dog or cat demonstrated more adaptive coping strategies (e.g., expressing emotions, seeking social support) than children with other types of pets and those without pets. This study further found that girls (but not boys) with a dog or cat reported the lowest rates of PTSD symptoms [16]. However, a similar study that investigated the influence of pet ownership on self-esteem in war-traumatized children did not find a difference between pet owners and non-pet owners [17]. Moreover, a study in survivors of the earthquake and tsunami in Japan in 2011 found that pet owners had higher rates of PTSD symptoms one month after the natural disaster, but significantly lower levels of PTSD symptoms after 4.4 years [18]. The authors explain their unexpected findings by higher levels of stress during initial trauma due to pet ownership, but superior long-term coping strategies.

Additionally, pet ownership was found to be beneficial in individuals that already developed a mental disorder (for a review see Brooks et al. [19]). Here, one has to differentiate between studies investigating the effects of psychiatric service dogs, who have been trained to perform commands relevant to patients’ psychological needs [20], and studies focusing on “regular” companion animals. While service dogs were found to have a positive effect on symptom levels and well-being [21,22,23,24], research on regular companion animals is rare. There is one study showing that PTSD patients also profit from regular companion animals: those that adopted a dog were found to feel calmer, less lonely, less depressed and less worried about their and their family’s safety [25]. Similar findings emerged in a sample of patients with treatment-resistant Major Depressive Disorder, showing that pet adoption led to less depressive symptoms as compared to a control group which did not adopt a pet [26].

Research suggested that pet ownership by itself does not impact human health, but that the type of pet and attachment to pets may play a moderating role. Dogs as “man’s best friend” received most attention in companion animal research, and dog ownership has been quite consistently related to a better physical health compared to owners of other types of pets and non-pet owners [27]. Furthermore, a recent study has shown that dog owners report higher levels of life satisfaction and happiness than cat owners [28]. In line, a recent longitudinal study in Japanese children found that dog ownership (but not cat ownership) at the age of ten was associated with better well-being two years later [29]. However, dogs also require more resources in terms of time (i.e., for care/walking) and costs, which may also result in an enhanced burden on dog owners as compared to owners of other pets [30]. Thus, the type of companion animal is an important factor that needs to be taken into account when analyzing the association of pet ownership and health.

Attachment to pets is another potential moderator of this relationship. Research suggests that humans are often strongly attached to their pets and sometimes report even stronger attachment to pets than to human family members [31,32,33]. Thus, recent research increasingly focused on the relationship between attachment to pets and mental health. However, these studies have yielded inconsistent results. While some found a substantial relationship between strong attachment to pets and lower degree of loneliness as well as depression [34,35], others did not find any association or even positive relationships between attachment to pets and loneliness as well as depression [36]. These inconsistent findings indicate that a stronger attachment to pets does not necessarily result in lower levels of psychopathological symptoms.

Even though research is not entirely conclusive, companion animals seem to have a buffering effect when individuals are exposed to critical life events, and preliminary evidence indicates that companion animals may be able to reduce existing psychopathological symptoms. However, to date little is known on how the relationship with a pet influences human health. One hypothesis is that companion animals may enhance levels of sense of coherence (SOC), the key component of the salutogenesis concept by Antonovsky [37,38]. SOC is considered as an adaptive dispositional orientation that enables coping with adverse experiences. The more an individual is able to understand and integrate (comprehensibility), to handle (manageability) and to make sense (meaningfulness) of life experience including stressors, the greater its potential to cope successfully with demands. SOC as a resistance factor is hypothesized to develop over time mainly during childhood and adolescence, depending on individual stress and bonding experiences as well as experiences relevant to self-esteem. It has been suggested that companion animals enhance SOC levels through enabling positive bonding experiences. To our knowledge, there is only one study investigating the relationship between dog ownership, SOC, and subjective evaluation of critical life events, which found—contrary to expectations—no differences in SOC levels between dog owners and others [39].

A health-benefitting factor closely related to SOC is trait-resilience, which can broadly be defined as the ability to adapt successfully in the face of adversity, trauma, or any other significant threat [40]. As described above, companion animals may serve as positive attachment figures. In line, pet ownership has been linked to better emotion regulation strategies and successful coping [16]. In turn, emotion regulation as well as the ability to seek social support have both been linked to resilience [41,42]. Thus, one may speculate that pet ownership is related to higher levels of trait-resilience. However, to our knowledge, there are no studies investigating the relationship between pet ownership or attachment to pets and trait-resilience.

Locus of control (LOC, [43]) is another concept discussed as a health-benefitting factor, which exhibits substantial conceptual overlap with SOC and trait-resilience. LOC assesses the degree to which individuals have the impression that events are controllable by their own actions (internal LOC) or predominantly depend on factors beyond their personal influence (external LOC). Previous research identified a stronger external LOC as a risk factor for PTSD symptoms [44], as a mediating factor between socioeconomic adversity and later depression [45], and as a robust correlate of psychopathological symptoms [46]. Research on LOC and pet ownership is scarce. One study found that individuals with physical disabilities (e.g., traumatic brain injuries) who received a service dog showed an enhanced internal LOC after six months, while a wait-list control group did not change in LOC [47]. However, to our knowledge there are no studies investigating differences in LOC between regular pet owners and non-pet owners.

Overall, research has shown that companion animals may enhance psychological health. Even though research is not entirely conclusive, these effects seem to be stronger for dog owners than for owners of other types of pets and are likely to depend on the strength of attachment. Furthermore, it has been hypothesized that companion animals may influence mental health through enhancement of health-benefitting factors such as SOC, trait-resilience, and LOC. However, hitherto, there are no studies analyzing differences between dog owners and non-dog owners with respect to psychopathological symptoms and health-benefitting factors. Moreover, little is known on the relationship between attachment to dogs, health-benefitting factors, and psychopathological symptom levels. The present cross-sectional survey aimed to address these gaps by assessing pet ownership (including type of pet), attachment to pets, health-benefitting factors, and psychopathological symptoms in a high-risk population consisting of different occupations. We expected dog owners to report higher levels of SOC, trait-resilience, stronger internal LOC and weaker external LOC as well as lower levels of psychopathological symptoms. Moreover, we hypothesized that differences between dog owners and non-dog owners in psychopathological symptom levels are explained by differences in health-benefitting factors. On an exploratory basis, we investigated the relationships between attachment to dogs and psychopathological symptom levels.

## 2. Materials and Methods

### 2.1. Sample Recruitment and Study Design

Respondents were recruited online by contacting different organizations and interest groups that represent specific high-risk occupations (e.g., trade unions for medical professions, police members, and firefighters). Moreover, advertisements were posted on specific websites addressing members of these occupations (e.g., Facebook groups sharing information on working at an intensive care unit). Additionally, respondents were asked to distribute the survey link at their individual workplaces and via social media. Sample recruitment for the 30 min online survey took place between February and November 2018. During that period, 750 respondents completed the survey. Of these, 170 respondents were excluded since they did not work in a field of interest. The final sample consisted of 223 medical staff members (i.e., nurses and practitioners), 257 police officers, and 100 firefighters. The study protocol was approved by the ethics committee of Saarland University (no. 16–2). According to the Declaration of Helsinki, all respondents gave written informed consent. Data used for this publication was also analyzed for a paper comparing different high-risk occupations with respect to psychopathological symptom levels and their association with health-benefitting factors, which will be published elsewhere.

### 2.2. Sample Characteristics

Of the 580 respondents of the total sample, 40.52% (*n* = 235) were female and 345 59.48% (*n* = 345) male. The mean age was 38.19 years (*SD* = ± 11.55 years). Across all occupations, the respondents reported a mean work experience of 16.68 years (*SD* = ± 11.54 years). Sixty percent of the respondents worked in shifts, with 50.51% working night and 19.82% working standby shifts. Forty-eight percent (*n* = 180) reported to own any type of pet. Of those, 51.79% (*n* = 145) were owners of a dog and 47.50% (*n* = 133) possessed a cat.

### 2.3. Measures

The online form started with 18 questions on socio-demographic information (i.e., gender, marital status, form of living, etc.), individual career and occupational characteristics (e.g., type of profession, work experience). Subsequently, respondents answered a set of standardized questionnaires on current psychopathological symptoms and health-benefitting factors. Moreover, respondents were asked if they own a pet, which type of pet they own, and they answered a standardized questionnaire assessing their emotional attachment to their most relevant pet. Due to time constraints, we were not able to assess additional information on pet ownership (e.g., favorite activities with the pet, age of pet, etc.).

#### 2.3.1. Health-Benefitting Factors

*Sense of coherence*. Two questionnaires were used to assess SOC. SOC as defined by Antonovsky [37,38] was measured using the German 13-item short version of the Antonovsky scales (SOC-13; [48]). SOC-13 uses a bipolar 7-point scale with verbal anchors at each side. Additionally, SOC-Revised (SOC-R; [49]) was assessed using a 13-item questionnaire developed by Bachem and Maercker [49]. For both SOC scales, higher scores indicate higher levels of SOC. In the present sample, SOC-13 demonstrated good internal consistency reflected in a Cronbach’s alpha (*α*) of 0.84. Findings on SOC-R will be part of another publication.

*Trait-resilience*. Resilience as a personality trait was assessed using the Resilience Scale 11 (RS-11; [50]). The RS-11 was developed as a short version of the 25-item Resilience Scale [51]. All items are rated on a bipolar 7-point scale. In the present study the internal consistency of RS-11 was good reflected in *α* = 0.90.

*Locus of control*. The concept of LOC was assessed using a 4-item brief scale for the assessment of control beliefs (IE-4; [52]). The instrument consists of two subscales (internal and external LOC) each comprising two items. All items are rated on a 5-point scale. As expected, items of each scale were correlated, *r*_internal_ = 0.36 and *r*_external_ = 0.37, and both scales were negatively correlated, *r* = -0.44.

#### 2.3.2. Psychopathological Symptom Burden

*General mental health problems*. The current burden on general mental health was assessed using the German version of the Brief Symptom Inventory [53]. The BSI is a 53-item self-report measure that assessed symptomatic distress using nine subscales. For the purpose of the current study, only the global severity index (GSI) was used to indicate general mental health problems. In the present sample, the GSI showed very good internal consistency reflected in *α* = 0.96.

*Posttraumatic stress symptoms*. Posttraumatic stress symptoms were measured using the German version of the Impact of Event Scale-Revised (IES-R; [54]). The IES-R assesses symptoms of re-experiencing, hyperarousal, and avoidance. The questionnaire consists of 22 items each rated on a 4-point scale. Item scores are transformed into a non-equidistant format (0, 1, 3, 5) resulting in a total score ranging from 0 to 110. Higher scores indicate a stronger burden caused by PTSD symptoms. The IES-R demonstrated good internal consistency in the current sample (α = 0.93).

*Burnout symptoms*. The German version of the Maslach Burnout Inventory–General Survey (MBI-GS; [55]) was used to assess symptoms of burnout in different occupations. The MBI consists of 22 items measuring three domains of burnout: emotional exhaustion (EE), depersonalization (DP), and personal accomplishment (PA). All items are rated on a 7-point scale. In the current sample internal consistencies were good (*α*_EE_ = 0.90, *α*_DP_ = 0.75, *α*_PA_ = 0.75).

*Attachment to pet*. The Lexington Attachment to Pets Scale [56] was used to assess emotional attachment to pets. The scale can be used for cats and dogs and consists of 23 items which are rated on a 4-point scale. Higher scores indicate a stronger attachment to pet. In the current sample, internal consistency was reflected in *α* = 0.93.

### 2.4. Data Collection, Data Aggregation and Analyses

All measures were collected via the online platform SoSci Survey [57]. Analyses were conducted using SPSS version 25 [58] and RStudio [59]. For the purpose of the current study, we compared dog owners to non-dog owners (i.e., owners of cats and other types of pets). Results for the comparison between dog/cat owners and non-dog/cat owners (i.e., owners of other types of pets and non-pet owners) are briefly summarized in the Results section and will be presented in detail as Appendix A.

Descriptive statistics were computed to illustrate sample characteristics in terms of frequencies, means (*M*), and standard deviations (*SD*). To assess differences between dog owners and non-dog owners, MANOVAs and *t*-tests for independent samples were conducted. To control for the effects of multiple testing, Bonferroni-Holm’s corrections were applied [60]. Moderator effects of age, gender, and form of living (alone vs. living together with a partner/family) were examined by means of ANOVAs, ANCOVAs and MANOVAs on an exploratory basis. Relevant moderating effects would be indicated by significant interaction terms. Moreover, Pearson’s bivariate correlation coefficients (*r*) were used to assess the relationships between SOC, trait-resilience, LOC, measures of psychopathological symptoms, and attachment to pets. We used a *z*-test for the comparison of correlations from independent samples to compare these correlations between dog owners and non-dog owners. Moreover, multiple regressions were conducted to determine the unique variance explained by each predictor variable that showed a significant bivariate correlation with the respective outcome variable. In order to account for the variance shared between health-benefitting factors and psychopathological symptoms, we conducted linear regression models including health-benefitting factors, which were shown to be significantly correlated with attachment to pets, and measures of psychopathological symptoms as predictors. Using these analyses, we were able to test if health-benefitting factors account for a significant unique amount of variance in attachment to pets or if the significant bivariate relationship simply reflects their well-known overlap [61,62] with psychopathological symptoms. Such a unique amount of variance would be reflected in a significant standardized regression weight (*β*), which corresponds to a significant change in explained variance (*∆R^2^*). Due to randomly missing data, degrees of freedom vary between analyses.

## 3. Results

### 3.1. Demographic Group Differences between Dog Owners and Non-Dog Owners

Sample characteristics of dog owners and non-dog owners are presented in Table 1. Dog owners and non-dog owners were not different regarding the proportion of women, *χ^2^* (1) = 0.70, *p* = 0.406, age, *t* (575) = 0.89, *p* = 0.371, and job experience, *t* (544) = 0.62, *p* = 0.530. However, dog owners were more likely to live together with a partner or family as compared to non-dog owners, *χ^2^* (1) = 4.45, *p* = 0.035. Moreover, they neither differed with respect to their frequency in working shifts, *χ^2^* (1) = 3.10, *p* = 0.078, nor to the proportion of those working night shifts or on standby duty.

### 3.2. Group Differences: Psychopathological Symptoms

*General mental health problems*. A *t*-test for independent samples with group (dog owner vs. non-dog owner) as an independent variable and GSI scores as dependent variable revealed no significant group difference, *t* (569) = 1.75, *p* = 0.081, *d* = 0.15. Moreover, an ANOVA revealed no moderating effects of gender, *F* (1,567) = 0.00, *p* = 0.999, *η^2^_p_* = 0.00, and age, *F* (1,564) = 0.51, *p* = 0.478, *η^2^_p_* = 0.00. Furthermore, living alone or together with a partner/family did not influence the results, *F* (1,567) = 0.09, *p* = 0.766, *η^2^_p_* = 0.00.

*Posttraumatic stress symptoms*. A *t*-test for independent samples with dog owner versus non-dog owner as independent variable and IES-R total scores as dependent variable did not show a significant group difference, *t* (496) = 1.06, *p* = 0.289, *d* = 0.10. As for general mental health problems, an ANOVA showed neither moderating effects of gender, *F* (1,494) = 0.71, *p* = 0.399, *η^2^_p_* = 0.00, age, *F* (1,491) = 0.81, *p* = 0.369, *η^2^_p_* = 0.00, nor form of living, *F* (1,494) = 0.07, *p* = 0.798, *η^2^_p_* = 0.00.

*Burnout symptoms*. A MANOVA with dog owner versus non-dog owners as between-subject factor and burnout symptoms (EE, DP, and PA) as dependent variable did not result in significant group differences, *F* (3,567) = 1.05, *p* = 0.370, *η^2^_p_* = 0.01. Moreover, MANOVAs found no significant moderator effects of gender, *F* (3,565) = 0.76, *p* = 0.516, *η^2^_p_* = 0.00, age, *F* (3,562) = 0.01, *p* = 0.999, *η^2^_p_* = 0.00, and form of living, *F* (3,565) = 1.17, *p* = 0.322, *η^2^_p_* = 0.01.

### 3.3. Group Differences: Health-Benefitting Factors

*Sense of coherence*. A *t*-test for independent samples with dog owner versus non-dog owner as group variable and SOC scores as dependent variable did not show significant group differences, *t* (578) = 1.69, *p* = 0.092, *d* = 0.14. Furthermore, three ANOVAs did not find moderator effects of gender, *F* (1,576) = 0.02, *p* = 0.885, *η^2^_p_* = 0.00, age, *F* (1,573) = 1.99, *p* = 0.159, *η^2^_p_* = 0.00, and form of living, *F* (1,576) = 1.14, *p* = 0.287, *η^2^_p_* = 0.00.

*Trait-resilience*. With respect to trait-resilience, a *t*-test for independent samples did not reveal significant differences between dog owners and non-dog owners, *t* (576) = 0.26, *p* = 0.791, *d* = 0.02. Moreover, ANOVAs did not find significant moderating effects; gender: *F* (1,574) = 0.57, *p* = 0.453, *η^2^_p_* = 0.00, age: *F* (1,571) = 1.39, *p* = 0.239, *η^2^_p_* = 0.00; form of living: *F* (1,574) = 0.06, *p* = 0.804, *η^2^_p_* = 0.00.

*Locus of control*. A MANOVA with dog owner versus non-dog owner as between-subject factor and external as well as internal LOC as dependent variables also did not result in a significant between group difference, *F* (2,577) = 1.27, *p* = 0.281, *η^2^_p_* = 0.004. Furthermore, MANOVAs did not find significant moderator effects; gender: *F* (2,575) = 0.12, *p* = 0.891, *η^2^_p_* = 0.00, age: *F* (2,572) = 2.31, *p* = 0.100, *η^2^_p_* = 0.01; form of living: *F* (2,575) = 0.26, *p* = 0.775, *η^2^_p_* = 0.00.

### 3.4. Summary of Findings on the Comparison between Dog/Cat Owners and Others

Running the same analyses with dog/cat owner as an independent variable revealed similar results (see Appendix A for details). Neither with respect to psychopathological symptoms (i.e., general mental health problems, PTSD symptoms, and burnout symptoms) nor health-benefitting factors (i.e., SOC, trait-resilience, and LOC), dog/cat owners and others were significantly different. Moreover, consistently across all analyses there were no moderator effects of age and form of living. However, women owning a dog were more likely to report more severe PTSD symptoms, while among non-dog/cat owners there were no gender difference in symptom levels between women and men.

### 3.5. Group Difference: Attachment to Pet

A *t*-test for independent samples with dog owner versus cat owner as independent variable and LAPS scores as dependent variable demonstrated a significant group difference, *t* (277) = 6.09, *p* < 0.001, *d* = 0.73. Dog owners showed a significant stronger attachment to their pet than cat owners. Moreover, moderator analyses showed that these differences were influenced by respondents’ age, *F* (1,274) = 4.65, *p* = 0.032, *η^2^_p_* = 0.02. Differences between attachment to dogs versus cats were larger in younger respondents and decreased as a function of age. However, there was no moderator effect of gender, *F* (1,275) = 3.78, *p* = 0.053, *η^2^_p_* = 0.01, and form of living, *F* (1,275) = 2.77, *p* = 0.097, *η^2^_p_* = 0.01.

### 3.6. Bivariate Correlations

Table 2 presents bivariate correlations between different measures of psychopathological symptoms, health-benefitting factors and emotional attachment to pets. All health-benefitting factors showed a significant relationship with measures of psychopathological symptom burden (all *p*s < 0.001). Attachment to pet was significantly correlated with general mental health problems, *r* = 0.27, *p* < 0.001, and PTSD symptoms, *r* = 0.29, *p* < 0.001. Moreover, EE was positively related to attachment to pets, *r* = 0.14, *p* = 0.016, while PA showed a negative association, *r* = -0.07, *p* < 0.001. With respect to health-benefitting factors, attachment to pets was associated negatively with SOC, *r* = -0.23, *p* < 0.001, and exhibited a positive relationship with external LOC, *r* = 0.15, *p* = 0.011.

These correlations were found to be numerically larger in dog owners as compared to cat owners. However, after applying Bonferroni-Holm corrections, correlations were not significantly different in size, *z* = 1.80, *p* = 0.324.

### 3.7. Multiple Regression Models

To further explore the association between attachment to pets and SOC and external LOC, which have shown to be significantly correlated with both psychopathological symptoms and attachment to pets, we conducted multiple regressions with attachment to pets as dependent variable and SOC/external LOC levels as predictor variable. Furthermore, we included different measures of psychopathological symptoms, i.e., general mental health problems, posttraumatic stress, and burnout symptoms, to control for their shared variance with attachment to pets. Significant unique amounts of variance in attachment to pets would be reflected in significant regression coefficients of SOC and external LOC.

Consistent across all measures of psychopathological symptoms (see Table 3), the relationship between attachment to pets and SOC was insignificant when accounting for general mental health problems, *β* (SOC) = -0.09, *t* (272) = -1.02, *p* = 0.310, and posttraumatic stress, *β* (SOC) = -0.12, *t* (232) = -1.75, *p* = 0.082, while these remained significant predictors of attachment to pets, *β* (GSI) = 0.21, *t* (272) = 2.42, *p* = 0.016; *β* (IES-R) = 0.23, *t* (232) = 3.30, *p* = 0.001. With respect to burnout symptoms, SOC remained a significant predictor of attachment to pets, *β* (SOC) = -0.25, *t* (271) = -3.26, *p* = 0.001, when analyzed in a joint model with EE, DP, and PA, while these did not account for significant amounts of variance, β (EE/DP/PA) ≤ |0.02|, *t* (271) ≤ 0.35, *p* ≥ 0.730. The same results were found for external LOC, when analyzed together with general mental health problems, *β* (LOC external = 0.05, t(272) = 0.80, *p* = 0.425, and posttraumatic stress, *β* (LOC external) = 0.08, t(232) = 1.13, *p* = 0.260, while neither external LOC, *β* (LOC external) = 0.13, *t* (271) = 1.95, *p* = 0.052, nor burnout symptoms, *β* (EE/DP/PA) ≤ 0.08, *t* (271) ≤ 1.03, *p* ≥ 0.302, showed incremental validity for attachment to pet, when analyzed in a joint model. Thus, except for burnout symptoms, the relationship between health-benefitting factors and attachment to pets seems to be explained by the association of health-benefitting factors and psychopathological symptoms.

## 4. Discussion

Up to date, there were no studies analyzing the relationship between pet ownership and psychopathological symptoms as well as health-benefitting factors in individuals working in high- risk occupations. The present cross-sectional survey aimed to address this research gap by assessing pet ownership (including type of pet), attachment to pets, health-benefitting factors (i.e., SOC, trait-resilience, and LOC), and psychopathological symptoms (i.e., general mental health problems, posttraumatic stress, and burnout symptoms). Contrary to our expectations, our results showed that dog owners did not differ from non-dog owners concerning health-benefitting factors and psychopathological symptoms. Dog owners were more strongly attached to their pets than cat owners. Additionally, a stronger attachment to pets was associated with more severe psychopathological symptoms and lower levels of health-benefitting factors. However, the relationship between attachment to pets and health-benefitting factors could be explained by their overlap with psychopathological symptom levels.

In contrast to our hypothesis dog owners (and pet owners in general) did not report higher levels of health benefitting factors (SOC, trait-resilience, and LOC) as compared to non-dog owners. We expected dog ownership to be associated with higher levels of health-benefitting factors because pet ownership has been associated with stronger social support and better emotion regulation strategies. However, up to date research on health-benefitting factors was scarce and the only study investigating the relationship between SOC and pet ownership did not find a significant association [39]. Moreover, the only study that assessed the relationship between LOC and pet ownership was conducted in individuals with physical disabilities, who showed an enhanced internal LOC six month after service dog adoption [47]. However, this sample may not be comparable to “regular” pet owners, because service dogs actually perform tasks for their owners, which in turn, may enhance their internal LOC, because they are less dependent on external help.

Dog owners also did not differ from non-dog owners regarding general psychopathological symptom levels, PTSD and burnout symptoms. These findings stand in contrast to our expectations and to other studies that found a relationship between dog/pet ownership and better mental health [7]. However, other studies also failed to find this relationship or even found a negative relationship between pet ownership and mental health [9]. One factor that has been hypothesized to account for these diverging findings is the moderating role of attachment to pets [31]. Our data showed that stronger attachment to pet was associated with lower levels of health-benefitting factors and more psychopathological symptoms. These findings are in line with previous research showing that stronger human–animal relationships are associated with increased reports of psychopathological symptoms [63,64,65,66]. However, they are inconsistent with findings that suggest either no association between living with a companion animal and well-being [67,68] or findings linking strong human–animal bonds with positive mental health outcomes [36]. One hypothesis - brought up by Müllersdorf et al. [69] - that may explain poorer mental health reported by pet owners strongly attached to their pet is that individuals who are predisposed to experience mental health problems are more likely to obtain a pet and bond to the pet strongly as some kind of self-help strategy. Pets are perceived as nondemanding, and it may be believed that they offer unconditional love to their owners. In our cross-sectional study, we did not assess when or for what reason the pet was acquired or how attachment to pet changed in the course of symptom development and thus, we are not able to test this hypothesis. However, studies looking at the potential benefits of recent dog adoption (in contrast to dog ownership) [70] showed that dog adoption was quite consistently associated with positive psychological outcomes such as reduced loneliness in healthy adults as well as patients with PTSD [25] and Major Depressive Disorder [26]. Nevertheless, more differentiated prospective studies are needed to investigate whether a stronger attachment to pets develops due to a higher psychopathological symptom burden or whether a stronger attachment to pets constitutes a risk factor for the development of psychopathology.

In our opinion, one misunderstanding in previous research is to equate strong attachment to pets with secure attachment. In our study, we used the Lexington Attachment to Pets Scale [56], a well-validated and widely used measure of attachment to pet with three subscales: general attachment, animal rights and welfare, and people substituting. Thus, the LAPS assesses the degree of attachment to pets, but does not provide any information about attachment style. Abundant research on mother–infant interactions has shown that there are different attachment styles and that secure attachment to others constitutes a protective factor for mental health [71]. However, it has not yet been investigated whether LAPS scores relate to human attachment styles. Other measures such as the Pet Attachment Questionnaire [72] are based on Bowlby’s attachment theory and may thus better assess a secure attachment to pets, which is supposed to exhibit a health-benefitting effect. Hence, future studies should incorporate this measure and focus on the question what degree or style of attachment to pet might relate to higher levels of well-being and health. Furthermore, it would have been very interesting and helpful to assess more information about the pets such as pets’ age, duration of stay with the owner, behavioural problems of the pet and time spent with the pet. These factors have been previously associated with attachment to pets but have to our knowledge not been investigated in the context of pet attachment and psychological well-being. Future studies should incorporate these measures in order to provide a more complete picture of the relationship between pet and owner [73,74].

A further explanation for our findings is that pet ownership and especially dog ownership requires resources in terms of time (i.e., for care/walking) and costs [30]. Attachment to pets has been linked to the time spent with the pet [75]. The sample of the present study consisted of individuals in high-risk occupations with many working shifts and overtime. Thus, the caring and time spent with the pet may also act as an additional stressor for individuals working at these demanding workplaces. Thus, more research is needed to disentangle the costs and benefits of pet ownership in specific populations.

In line with previous studies [76,77], in our sample dog owners were more strongly attached to their pets than cat-owners. However, there are also studies reporting no differences between dog and cat owners (with dog owners and cat owners scoring significantly higher than owners of other types of pets). The often-reported closer attachment to dogs may also reflect a methodological confound, since some items of questionnaires on attachment to pets only describe activities typical of dogs. When these items are removed, dog owners and cat owners were found to report similar levels of attachment on the Comfort from Companion Animals Scale (CCAS, [76]). In a similar study measuring only the emotional aspect of attachment to pets, Winefield and colleagues [77] reported no differences between dog and cat owners. Thus, the stronger bond to dogs might be biased by pet attachment instruments being designed to measure attachment to dogs instead of attachment to pets in general.

## 5. Conclusions

Despite its limitations, this was (to our knowledge) the first study to investigate the relationship between pet ownership, health benefitting factors and psychopathological symptoms in individuals working in high-risk occupations. Our study showed that dog owners (as well as dog/cat owners) compared to those owning other types of pets and non-pet owners were not different with respect to health-benefitting factors and psychopathological symptoms. A stronger attachment to pets was associated with higher levels in psychopathological symptoms and lower levels of health-benefitting factors. However, further analyses showed the relationship between health-benefitting factors and attachment to pets could be explained by the association of health-benefitting factors and psychopathological symptoms.

Thus, our data contribute to the growing body of evidence questioning the general health-benefitting effect of pet ownership. More differentiated prospective studies are needed to determine whether a stronger attachment to pets develops due to higher psychopathological symptom burden or whether a stronger attachment to pets constitutes a risk factor for the development of psychopathology in high-risk occupations and other groups of interest.

## Figures and Tables

**Table 1 ijerph-17-02562-t001:** Sample characteristics for dog owners and non-dog owners.

	Dog Owners	Non-Dog Owners	*χ^2^/t* (df)	*p*
**Sex (% women)**	43.40	39.05	*χ^2^* (1) = 0.70	0.406
**Age (in years)**	38.94	37.94	*t* (575) = 0.89	0.371
	(10.76)	(11.81)		
**Form of living (alone %)**	17.10	28.30	*χ^2^* (1) = 4.45	0.035
**Job experience (in years)**	17.21	16.50	*t* (544) = 0.62	0.530
	(10.80)	(11.78)		
**Shift work (%)**	66.20	57.90	*χ^2^* (1) = 3.10	0.078
**Night shifts (% of those working shifts)**	84.40	84.10	*χ^2^* (1) = 0.00	0.955
**Standby duty (% of those working shifts)**	30.20	34.10	*χ^2^* (1) = 0.48	0.487

*Note*. df = degree of freedom. Numbers in brackets indicate standard deviations or degrees of freedom.

**Table 2 ijerph-17-02562-t002:** Pearson correlations between psychopathological symptoms, health-benefitting factors and emotional attachment to the pet.

	1		2		3		4		5		6		7		8		9		10	
GSI (1)																				
IES-R total (2)	0.53	***																		
MBI_EE_ (3)	0.59	***	0.45	***																
MBI_DP_ (4)	0.37	***	0.27	***	0.58	***														
MBI_PA_ (5)	−0.32	***	−0.3	***	−0.25	***	−0.2	***												
SOC-13 (6)	−0.73	***	−0.49	***	−0.59	***	−0.44	***	0.42	***										
trait-resilience (7)	−0.52	***	−0.34	***	−0.4	***	−0.23	***	0.48	***	0.54	***								
LOC_internal_ (8)	−0.38	***	−0.35	***	−0.42	***	−0.24	***	0.33	***	0.5	***	0.45	***						
LOC_external_ (9)	0.43	***	0.38	***	0.41	***	0.24	***	−0.18	***	−0.53	***	−0.31	***	−0.44	***				
LAPS (10)	0.27	***	0.29	***	0.14	*	0.1		−0.07	***	−0.23	***	−0.04		−0.03		0.15	*		
Dog owners	0.34	***	0.36	***	0.14		0.13		−0.07		−0.024	***	0.01		0.08		0.13			
Cat owners	0.14		0.18	*	0.11		−0.001		0.02		−0.016		−0.05		−0.07		0.13			
*Z*	1.8																			
*p* _adjusted_	0.324																			

* *p* < 0.05, *** *p* < 0.001. *Note*. GSI = global severity index; IES-R total = Impact of Event Scale-Revised total score; MBI_EE/DP/PA_ = Maslach Burnout Inventory emotional exhaustion (EE)/depersonalization (DP) /personal accomplishment (PA); SOC = sense of coherence; LOC = locus of control; LAPS = Lexington Attachment to Pets Scale.

**Table 3 ijerph-17-02562-t003:** Multiple regression results for the prediction of attachment to pets (LAPS total score).

	B	SE B	*β*	*t*	*p*
Sense of Coherence					
GSI	0.04	0.02	0.21	2.42	0.016
SOC	−0.01	0.00	−0.09	−1.20	0.310
IES-R total	0.23	0.07	0.23	3.30	0.001
SOC	−0.12	0.07	−0.12	−1.75	0.082
MBI_emotional exhaustion_	−0.02	0.08	−0.002	−0.02	0.985
MBI_depersonalization_	0.01	0.07	0.01	0.08	0.934
MBI_personal_ _accomplishment_	0.02	0.07	0.002	0.35	0.730
SOC	−0.25	0.08	−0.25	−3.26	0.001
External locus of control					
GSI	0.04	0.01	0.25	3.89	<0.001
LOC_external_	0.05	0.06	0.05	0.80	0.425
IES-R total	0.26	0.07	0.26	3.89	<0.001
LOC_external_	0.08	0.07	0.08	1.13	0.260
MBI_emotional exhaustion_	0.07	0.08	0.08	1.03	0.302
MBI_depersonalization_	0.02	0.07	0.02	0.28	0.778
MBI_personal_ _accomplishment_	−0.03	0.06	−0.03	−0.56	0.573
LOC_external_	0.13	0.07	0.13	1.95	0.052

*Note*. GSI = global severity index; IES-R total = Impact of Event Scale-Revised total score; MBI = Maslach Burnout Inventory; SOC = sense of coherence; LOC = locus of control; LAPS = Lexington Attachment to Pets Scale.

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
