# Peer review of "The Relationship between Dog Ownership, Psychopathological Symptoms and Health-Benefitting Factors in Occupations at Risk for Traumatization"

_ijerph, 2020, doi:10.3390/ijerph17072562_

Round 1
Reviewer 1 Report
The authors report a well thought study about the relationship between pet ownership and mental health (psychopathological symptoms and health-benefitting factors, e.g. trait resilience markers) in individuals working in high-risk occupations such as medical staff, police officers and firefighters using a cross-sectional online survey. Interestingly, the results showed no difference in psychopathological symptoms and health-benefitting factors between dog owners and non-dog owners. The authors discuss the role of attachment as well as additional effort for pet care or higher previous psychopathological symptoms in dog owners as potential reasons. In my opinion, these are important results in the research field of pets and mental health. Usually, pet-ownership has a health-benefitting effect, but these results underline the great importance to understand mechanisms how pet-ownership might contribute to mental health. The results point out important directions for further research and for the development of new therapeutical/ prevention strategies to increase mental resilience, especially in high risk groups. Overall, the manuscript is well written and I only have some remarks:
- Because it was a bit difficult to follow the results section, I would recommend to describe the analyses (and the analyzed groups) more in detail in the section „Data collection and Analyses“ (2.4). In addition, an additional table with the results of the multiple regression models (3.7) could be helpful.
- Please specify in your line of reasoning and your hypothesis if the benefical effect of pet ownership is expected for dogs or animals in general.
Author Response
We thank you very much for your time and effort in reviewing our manuscript. Our comments helped a lot when preparing a revised version and we believe that your input was very helpful to improve our work. Please find below our detailed response to our comments.
R 1.1. Because it was a bit difficult to follow the results section, I would recommend to describe the analyses (and the analyzed groups) more in detail in the section „Data collection and Analyses“ (2.4). In addition, an additional table with the results of the multiple regression models (3.7) could be helpful.
We thank you for this helpful suggestion and added information on the analyses and their rationale. We now described the analyses in greater detail in the section “Data Collection, Data Aggregation and Analyses”. In detail, we write (see p. 5, lines 251 following):
In order to account for the variance shared between health-benefitting factors and psychopathological symptoms, we conducted linear regression models including health-benefitting factors, which were shown to be significantly correlated with attachment to pets, and measures of psychopathological symptoms as predictors. Using these analyses, we were able to test if health-benefitting factors account for a significant unique amount of variance in attachment to pets or if the significant bivariate relationship simply reflects their well-known overlap [e.g., 61,62] with psychopathological symptoms. Such a unique amount of variance would be reflected in a significant standardized regression weight (β), which corresponds to a significant change in explained variance (∆R2).
Moreover, we added a table including the results of the multiple regression models to our Results section (see p. 8, line 361 following).
Table 3. Multiple regression results for the prediction of attachment to pets (LAPS total score)
|
|
B |
SE B |
β |
t |
p |
|
Sense of coherence |
|
|
|
|
|
|
GSI |
0.04 |
0.02 |
.21 |
2.421 |
.016 |
|
SOC |
-0.01 |
0.00 |
-.13 |
-2.20 |
.029 |
|
IES-R total |
0.23 |
0.07 |
.23 |
3.30 |
.001 |
|
SOC |
-0.12 |
0.07 |
-.12 |
-1.75 |
.082 |
|
MBIemotional exhaustion |
-0.02 |
0.08 |
-.002 |
-0.02 |
.985 |
|
MBIdepersonalization |
0.01 |
0.07 |
.01 |
0.08 |
.934 |
|
MBIpersonal accomplishment |
0.02 |
0.07 |
.002 |
0.35 |
.730 |
|
SOC |
-0.25 |
0.08 |
-.25 |
-3.26 |
.001 |
|
External locus of control |
|
|
|
|
|
|
GSI |
0.04 |
0.01 |
.25 |
3.89 |
< .001 |
|
LOCexternal |
0.05 |
0.06 |
.05 |
0.80 |
.425 |
|
IES-R total |
0.26 |
0.07 |
.26 |
3.89 |
< .001 |
|
LOCexternal |
0.08 |
0.07 |
.08 |
1.13 |
.260 |
|
MBIemotional exhaustion |
0.07 |
0.08 |
.08 |
1.03 |
.302 |
|
MBIdepersonalization |
0.02 |
0.07 |
.02 |
0.28 |
.778 |
|
MBIpersonal accomplishment |
-0.03 |
0.06 |
-.03 |
-0.56 |
.573 |
|
LOCexternal |
0.13 |
0.07 |
.13 |
1.95 |
.052 |
Note. GSI = general severity index; IES-R total = Impact of Event Scale-Revised total score; MBI = Maslach Burnout Inventory; SOC = sense of coherence; LOC = locus of control; LAPS = Lexington Attachment to Pets Scale.
R 1.2 Please specify in your line of reasoning and your hypothesis if the beneficial effect of pet ownership is expected for dogs or animals in general.
Thank you for bringing this lack of clarity to our attention. We agree with you that effects of pet ownership in general and dog ownership should be differentiated in the most stringent way currently possible. When rereading our manuscript, we realized that we were not very consistent in the use of “beneficial effect of pet ownership“ and “beneficial effect of dog ownership“ when developing our hypotheses in the Introduction section. Even though the literature is not totally conclusive, we expected the beneficial effect to be stronger for dog owners as compared to owners of other types of pets and non-pet owners. In order to specify our line of reasoning and hypotheses, we also changed the title of the manuscript into “The relationship between dog ownership, psychopathological symptoms and health-benefitting factors in occupations at risk for traumatization”. Furthermore, we now consistently relate to dog ownership instead of pet ownership, when describing our hypotheses and research question (see p. 3, lines 139 following). Due to the current state of research, in our review of the literature, we still also refer to beneficial effects of pet ownership in general.
Reviewer 2 Report
The paper is of interest and is relevant to the human-animal bond research. It is consistent and well wrote. It would be interesting to have more descriptive data about pets: age, sex, time with the family, type of activities owners share with their pets. These variables could explain better the benefits of pets on the dependent variables. It is not just about owning or non owning a pet, it is the type of relationship with the pet that makes the difference.
If authors do not have data to include this information in results they should comment about it in discussion.
Author Response
We thank you very much for your time and effort in reviewing our manuscript. Your comments on our manuscript were very helpful and inspired us to conduct additional moderator analyses, which add relevant information to our findings. Please find below our detailed response to our comments.
R 2.1 It would be interesting to have more descriptive data about pets: age, sex, time with the family, type of activities owners share with their pets. These variables could explain better the benefits of pets on the dependent variables. It is not just about owning or non owning a pet, it is the type of relationship with the pet that makes the difference. If authors do not have data to include this information in results they should comment about it in discussion.
We agree with you that more descriptive data on pets would have been very interesting and relevant additional information. In order to capture also qualitative aspects of pet ownership, we included the assessment of attachment to pets using the Lexington Attachment to Pets Scale in our study. Nevertheless, we agree with you that additional information on pets’ age, time with the family, active time spent with the pets etc. would have given a more complete picture of the relationship with the pet. Due to time constraints (i.e., respondents were not payed for their participation and the online survey should not take longer than 30 minutes), we were not able to include this information into our questionnaire. We now added this point to the Discussion section, in detail, we write (p. 10, lines 431 following):
Furthermore, it would have been very interesting and helpful to assess more information about the pets such as pets’ age, duration of stay with the owner, behavioural problems of the pet and time spent with the pet. These factors have been previously associated with attachment to pets but have to our knowledge not been investigated in the context of pet attachment and psychological well-being. Future studies should incorporate these measures in order to provide a more complete picture of the relationship between pet and owner [73,74].
However, we did assess more details on pet owner characteristics [such as gender, age, and form of living (alone vs. with partner or family)], which have also been previously associated with attachment to pets, and we now include these measures as potential moderators. We believe that these analyses do not replace details on the relationship between owners and their pet, but they may provide further information about potentially relevant circumstances. To sum, we did not find significant moderator effects for these variables. In detail, we added the following details to our Methods and Results section:
Methods section (p. 5, line 242 following)
Moderator effects of age, gender, and form of living (alone vs. living together with a partner/family) were examined by means of ANOVAs, ANCOVAs and MANOVAs on an exploratory basis. Relevant moderating effects would be indicated by significant interaction terms.
Results section (p. 6, line 261 following):
3.2. Group Differences: Psychopathological Symptoms
General mental health problems. A t-test for independent samples with group (dog owner vs. non-dog owner) as independent variable and GSI scores as dependent variable revealed no significant group difference, t(569) = 1.75, p = .081, d = 0.15. Moreover, an ANOVA revealed no moderating effects of gender, F(1, 567) = 0.00, p = .999, η2p = .00 , and age, F(1, 564) = 0.51, p = .478, η2p = .00. Furthermore, living alone or together with a partner/family did not influence the results, F(1, 567) = 0.09, p = .766, η2p = .00.
Posttraumatic-stress symptoms. A t-test for independent samples with dog owner versus non-dog owner as independent variable and IES-R total scores as dependent variable did not show a significant group difference, t(496) = 1.06, p = .289, d = 0.10. As for general mental health problems, an ANOVA showed neither moderating effects of gender, F(1, 494) = 0.71, p = .399, η2p = .00, age, F(1, 491) = 0.81, p = .369, η2p = .00, nor form of living, F(1, 494) = 0.07, p = .798, η2p = .00.
Burnout symptoms. A MANOVA with dog owner versus non-dog owners as between-subject factor and burnout symptoms (EE, DP, and PA) as dependent variable did not result in significant group differences, F(3, 567) = 1.05, p = .370, η2p = .01. Moreover, MANOVAs found no significant moderator effects of gender, F(3, 565) = 0.76, p = .516, η2p = .00, age, F(3, 562) = 0.01, p = .999, η2p = .00, and form of living, F(3, 565) = 1.17, p= .322, η2p = .01.
3.3. Group Differences: Health-Benefitting Factors
Sense of coherence. A t-test for independent samples with dog owner versus non-dog owner as group variable and SOC scores as dependent variable did not show significant group differences, t(578) = 1.69, p = .092, d = 0.14. Furthermore, three ANOVAs did not find moderator effects of gender, F(1, 576) = 0.02, p = .885, η2p = .00, age, F(1, 573) = 1.99, p = .159, η2p = .00, and form of living, F(1, 576) = 1.14, p = .287, η2p = .00.
Trait-resilience. With respect to trait-resilience, a t-test for independent samples did not reveal significant differences between dog owners and non-dog owners, t(576) = 0.26, p = .791, d = 0.02. Moreover, ANOVAs did not find significant moderating effects; gender: F(1, 574) = 0.57, p = .453, η2p = .00, age: F(1, 571) = 1.39, p = .239, η2p = .00, form of living: F(1, 574) = 0.06, p = .804, η2p = .00.
Locus of control. A MANOVA with dog owner versus non-dog owner as between-subject factor and external as well as internal LOC as dependent variables also did not result in a significant between group difference, F(2, 577) = 1.27, p = .281, η2p = .004. Furthermore, MANOVAs did not find significant moderator effects; gender: F(2, 575) = 0.12, p = .891, η2p = .00, age: F(2, 572) = 2.31, p = .100, η2p = .01, form of living: F(2, 575) = 0.26, p = .775, η2p = .00.
3.4. Summary of Findings on the Comparison Between Dog/Cat Owners and Others
Running the same analyses with dog/cat owner as an independent variable, revealed similar results (see Supplementary Material A for details). Neither with respect to psychopathological symptoms (i.e., general mental health problems, PTSD symptoms, and burnout symptoms) nor health-benefitting factors (i.e., SOC, trait-resilience, and LOC) dog/cat owners and others were significantly different (see Supplementary Material A for detailed results). Moreover, consistently across all analyses, there were no moderator effects of age and form of living. However, women owning a dog were more likely to report more severe PTSD symptoms, while among non-dog/cat owners there were no gender difference in symptom levels between women and men.
Reviewer 3 Report
The relationship between pet ownership, psychopathological symptoms and health-benefitting factors in occupations at risk for traumatization (ijerph-753665)
Journal: International Journal of Environmental Research and Public Health
Study design employs a cross-sectional online survey of 580 medical staff, police officers, and firefighters. Authors correctly emphasize that these occupations lead to “work-related traumatic events repeatedly resulting in a cumulative burden” that increases the risk for burnout, depression and posttraumatic stress disorder. Due to lack of evidence regarding the effect of pet ownership on traumatic life events, this target population was chosen to measure how pet ownership affects the response of these workers to stressors and critical life events. Regular companion animals were studied as opposed to service animals. The main hypothesis was that dog owners in these occupations would report higher levels of SOC, trait-resilience, stronger internal LOC, weaker external LOC, and lower levels of psychopathological symptoms compared to non-dog owners. Analysis included measuring the Pearson’s bivariate correlation coefficients between SOC, trait-resilience, LOC, measures of psychopathological symptoms, and attachment to pets; then significant variables were entered into multiple regression. Dog owners had higher LAPS scores for attachment to their pet than cat owners. Attachment to pet was significantly correlated with general mental health problems. Stronger attachment to pets was associated with more severe psychopathological symptoms. Dog owners (and pet owners in general) did not report higher levels of SOC, trait-resilience, and LOC compared to non-dog owners. Overall results do not suggest that pet ownership had any benefit in this context.
The Discussion is well balanced, informed and raises many important issues for further study. Most importantly, the Discussion addresses Müllersdorf’s hypothesis that individuals with mental health problems are more likely to obtain a pet and bond to the pet strongly as a “self-help strategy” and benefit from a pet’s unconditional love. A main limitation of cross-sectional design in this study is that it did not determine why the pet was obtained, when the pet was acquired or the temporal relationship between the owner’s psychopathological symptoms evolution and the time the pet was obtained. Thus the study could not test this hypothesis. But authors should mention the need for longitudinal studies to fully address this issue. Thus in the Conclusion the authros are correct in stating that “prospective studies are needed to determine whether a stronger attachment to pets develops due to higher psychopathological symptom burden or whether a stronger attachment to pets constitutes a risk factor for the development of psychopathology in high-risk occupations and other groups of interest.”
Question: The authors stated a few times that ‘the relationship between health-benefitting factors and attachment to pets could be explained by the association of health-benefitting factors and psychopathological symptoms’. Does that imply a collinear relationship? Would collinearity a better way to describe this relationship?
Other comments:
Page 9 line355 “dog adoption was quiet consistently” – appears to be a typo and should be changed to “quite” or deleted
Author Response
We thank you very much for your time and effort in reviewing our manuscript. Your comments were very helpful when preparing a revised version of our manuscript. Please find below our detailed response to our comments.
R 3.1 A main limitation of cross-sectional design in this study is that it did not determine why the pet was obtained, when the pet was acquired or the temporal relationship between the owner’s psychopathological symptoms evolution and the time the pet was obtained. Thus, the study could not test this hypothesis. But authors should mention the need for longitudinal studies to fully address this issue. Thus in the Conclusion the authors are correct in stating that “prospective studies are needed to determine whether a stronger attachment to pets develops due to higher psychopathological symptom burden or whether a stronger attachment to pets constitutes a risk factor for the development of psychopathology in high-risk occupations and other groups of interest.”
We strongly agree with you that this is a main limitation of our cross-sectional design and we now elaborated more on the need of longitudinal studies to further explore the relationship between pet acquisition, pet attachment and psychopathological symptoms over longer periods. In detail, we write (p. 10, line 411 and following):
In our cross-sectional study, we did not assess when or for what reason the pet was acquired or how attachment to pet changed in the course of symptom development and thus, we are not able to test this hypothesis. However, studies looking at the potential benefits of recent dog adoption (in contrast to dog ownership) [70]showed that dog adoption was quite consistently associated with positive psychological outcomes such as reduced loneliness in healthy adults as well as patients with PTSD [25] and Major Depressive Disorder [26]. Nevertheless, more differentiated prospective studies are needed to investigate whether a stronger attachment to pets develops due to higher psychopathological symptom burden or whether a stronger attachment to pets constitutes a risk factor for the development of psychopathology.
R 3.2 The authors stated a few times that ‘the relationship between health-benefitting factors and attachment to pets could be explained by the association of health-benefitting factors and psychopathological symptoms’. Does that imply a collinear relationship? Would collinearity a better way to describe this relationship?
Thank you for bringing this lack of clarity to our attention. Based on our multiple regression analyses, we aimed to investigate if the variance that is accounted for by health-benefitting factors in attachment to pets is equal to the variance they commonly share with psychopathological symptoms. Thus, we were not interested if these factors are collinear per se (which would require much larger bivariate correlations between health-benefitting factors and psychopathological symptoms), but if the variance that they account for in attachment for pets is redundant and thus, probably an epiphenomena of both being associated with symptom levels (or vice versa).
Moreover, we calculated collinearity statistics for all analyses included in our manuscript. For example, when we predicted attachment to pets based on PTSD symptoms and SOC, all condition indices were ≤ 16.78 (collinearity is associated with scores ≥ 30). However, in the same analysis, the health-benefitting factor SOC, which was correlated significantly with attachment to pets, r = -.23, p < .001, was no longer a significant predictor, βSOC = -.12, t(232) = -1.75, p = .082. Thus, variance that SOC levels explain in attachment to pets is already accounted for by the overlap between PTSD symptoms and attachment to pets (while PTSD symptoms explain an incremental amount of variance above SOC).
In order to further explain the rationale of our analyses to the reader, we included a more detailed description in the Method section. In detail, we write (see p. 5, lines 251 following):
In order to account for the variance shared between health-benefitting factors and psychopathological symptoms, we conducted linear regression models including health-benefitting factors, which were shown to be significantly correlated with attachment to pets, and measures of psychopathological symptoms as predictors. Using these analyses, we were able to test if health-benefitting factors account for a significant unique amount of variance in attachment to pets or if the significant bivariate relationship simply reflects their well-known overlap [e.g., 61,62] with psychopathological symptoms. Such a unique amount of variance would be reflected in a significant standardized regression weight (β), which corresponds to a significant change in explained variance (∆R2).
R 2.3 Page 9 line355 “dog adoption was quiet consistently” – appears to be a typo and should be changed to “quite” or deleted
Thank you very much for making us aware of this misspelling. This has been corrected in the revised version of our manuscript.
Round 2
Reviewer 1 Report
The authors did a great job in revising the manuscript. All my points have been adressed in a satisfactory way. I have no further concerns.